# WHEN VALIDITY ISN'T ENOUGH: RELIABILITY GAPS IN MOLECULAR GENERATION & KRAS CASE STUDY

## ABSTRACT

Molecule generation remains a core challenge in computational chemistry. Practical use of generative models is complicated by strict chemical, structural, and biological constraints: candidate compounds must satisfy physicochemical bounds, avoid reactive or toxic substructures, be synthesizable, and plausibly bind a target. We are the first to perform such comprehensive analysis of modern molecule generators via the Five-Stage Filtering Pipeline, a target-agnostic, practice-oriented benchmark for evaluating *de novo* generators using the following stages: (i) physicochemical descriptors; (ii) structural alerts; (iii) synthesis feasibility; (iv) docking and binding affinity estimation; and (v) blind medicinal chemist review. We compare 18 generators across three families (unconditional, ligand-based, and protein-based), and to make it practically relevant, apply the pipeline to KRAS G12D switch-II pocket for conditional design case study. Less than $1\%$ of molecules pass all stages, exposing a gap between high scores on standard generative metrics and practical medicinal chemistry usage. We release our benchmark, and code to enable reproducible evaluation and to focus future model development on practically useful chemical space.

## 1 INTRODUCTION

One of the central challenges of biomedicine of the 21st-century is to prevent and treat complex diseases, improve population health, and extend human longevity (Hood et al., 2004; Kirkwood, 2005; Murray et al., 2012). Early drug discovery addresses this challenge through a staged pipeline: (i) identify a biological *target* associated with a disease; (ii) identify a surface, known as a *pocket*, using crystallography or pocket prediction software; and (iii) design a small molecule, known as a *ligand*, that binds and modulates the *target*. Despite decades of progress, identifying high-quality *ligands* remains labor-intensive, time-consuming, and expensive (Paul et al., 2010).

Machine learning is reshaping this landscape by accelerating design–make–test cycles for *de novo* molecular generation (Zhavoronkov et al., 2019). To be considered a viable drug candidate, a molecule must satisfy multiple criteria (Hughes et al., 2011; Waring et al., 2015; Lipinski, 2004). First, physicochemical properties must fall within reasonable bounds, e.g., limited rotatable bonds and polar surface area controlling permeability and oral exposure (Veber et al., 2002). Second, reactive chemotypes and structural alerts associated with toxicity should be removed or flagged, e.g., PAINS substructures filter out potential assay-interference compounds (Baell & Holloway, 2010; Huggins et al., 2011; Sushko et al., 2012). Third, the candidate should be practically synthesizable, which is estimated with heuristic scores and retrosynthesis planning (Ertl & Schuffenhauer, 2009; Coley et al., 2017; Genheden et al., 2020).

However, recent studies show that high scores on popular generative benchmarks often fail to translate into synthesizable, medicinally plausible compounds (Bodenreider et al., 2021). Efficient filtering of generated molecules is therefore essential prior to hit identification and lead optimization (Schneider & Fechner, 2010; Hughes et al., 2011). Without rigorous filtration, computational and experimental resources are wasted on non-viable candidates; with it, molecules meeting chemical, medicinal, and task-dependent criteria proceed further, improving success rates and reducing costs.

To the best of our knowledge, we introduce the first comprehensive, practice-oriented benchmark for evaluating *de novo* molecular generators under realistic medicinal chemistry constraints. We compare three generator families - (i) unconditional generators, (ii) ligand-based models, and

(iii) structure-aware protein-based models - and investigate whether they generate molecules that pass a realistic multi-stage filter cascade. As a biologically relevant case study, we focus on KRAS G12D mutant, for which no approved inhibitors exist; by contrast, KRAS G12C mutant has FDA approved drugs - sotorasib (Blair, 2021; Hong et al., 2020) and adagrasib (Jänne et al., 2022; Canon et al., 2019). For protein-based tasks, we focus generation on the switch-II pocket of KRAS G12D using PDB 7EW9 (PDB ID: pdb_00007ew9), a GDP-bound KRAS G12D structure in complex with TH-Z816. For ligand-based tasks, we condition generation on known KRAS G12D inhibitors (Ghazi Vakili et al., 2025).

Our contributions are:

- We propose the Five-Stage Filtering Pipeline for molecule evaluation with: coarse physico-chemical descriptors, medicinal chemistry alerts, synthetic feasibility, docking and binding affinity estimation, and blind medicinal chemistry scoring.

- We propose a standardized target-agnostic filtering and evaluation process applicable to *unconditional*, *ligand-based*, and *protein-based* generators; as a case study, we employ a unified evaluation via docking and binding affinity estimation against KRAS G12D.

- We show that under our pipeline, only a small fraction (less than $1\%$) of generated molecules pass all filters and remain applicable for future work.

- We demonstrate that unconditional models show the highest pass rates among other families; ligand-based models more often violate coarse descriptor bounds; and protein-based models show the lowest pass rate.

Overall, our benchmark prioritizes stress-testing diverse molecular generators against constraints that matter in drug discovery settings, and shifts evaluation toward actionable chemical space. The protocol is extensible to new targets by swapping the pocket definition and ligand sets while keeping the filter cascade unchanged, enabling reproducible comparisons.

## 2 RELATED WORK

We categorize molecular generators by generative strategy and architecture because both impose distinct inductive biases - validity and grammar errors for strings, geometry handling for 3D models, pocket alignment for pocket-based models (David et al., 2020; Bilodeau et al., 2022). Table 1 summarizes the mapping and models are described below.

Table 1: Taxonomy of molecular generators considered in our benchmark, by architecture (rows) and generative strategy (columns)

| Architecture /Model Type | UNCONDITIONAL | LIGAND-BASED | PROTEIN-BASED |
|---|---|---|---|
| Genetic Algorithm | — | MolFinder (Kwon & Lee, 2021) | — |
| Variational Autoencoder | HierGraphVAE (Jin et al., 2020) JT-VAE (Jin et al., 2018) MoLeR (Maziarz et al., 2021) | GENTRL (Zhavoronkov et al., 2019) | — |
| Autoregressive | MolGPT (Bagal et al., 2021) | GCPG (Zou et al., 2025) PGMG (Zhu et al., 2023) REINVENT4 (Loeffler et al., 2024) | Dragonfly (Atz et al., 2024) Pocket2Mol (Peng et al., 2022) ResGen (Zhang et al., 2023) |
| Diffusion | E(3)DM (Hoogeboom et al., 2022) TGM-DLM (Gong et al., 2024) | — | DiffSBDD (Schneuing et al., 2024) ProtoBind-Diff (Mistryukova et al., 2025) TargetDiff (Guan et al., 2023) |
| Flow matching | — | — | DrugFlow (Schneuing et al., 2025) |

**Genetic algorithm (GA)** GA is a heuristic optimizer that evolves molecules via crossover and mutation operations. **MolFinder** (Kwon & Lee, 2021) applies Conformational Space Annealing to SMILES (Weininger, 1988), and requires no generative model to pretrain for ligand-based design.

**Variational autoencoder (VAE)** VAE models learn a latent distribution over chemical space with an encoder-decoder pair optimized via the ELBO (Kingma & Welling, 2013). **JT-VAE** (Jin et al., 2018), **HierGraphVAE** (Jin et al., 2020), **MoLeR** (Maziarz et al., 2021) operate on graphs with scaffold-aware decoders. They typically yield high validity and diversity, but as unconditional generators, they do not ensure target relevance. **GENTRL** (Zhavoronkov et al., 2019) is a string VAE

with Reinforcement Learning (RL) fine-tuning, which generates molecules with high similarity to target molecules.

**Autoregressive models** String models factorize the sequence likelihood as $\prod_i P(t_i \mid t_{<i})$. **Mol-GPT** (Bagal et al., 2021) is a decoder-only Transformer, with no target protein or ligands hints. REINVENT (Olivecrona et al., 2017) fine-tunes a SMILES *Prior* into an *Agent* via policy gradient to maximize a scoring function. **REINVENT4** (Loeffler et al., 2024) generalizes REINVENT to RNN or Transformer priors, combining transfer learning, curriculum learning, or RL with a multi-component scoring system for goal-directed design.

For structure-based design, 3D autoregressive models condition on a pocket $P$ and learn conditional likelihood of a molecule $M$ as $p_\theta(M \mid P) = \prod_{t=1}^{T} p_\theta(z_t \mid z_{<t}, P)$, where each step $z_t$ adds atom, bond, or coordinates. **Pocket2Mol** (Peng et al., 2022), **ResGen** (Zhang et al., 2023), and **Dragonfly** (Atz et al., 2024) encode pocket geometry with SE(3) or E(3) equivariant encoders, and decode pocket-aware ligands.

Pharmacophore-based models use $c$ as a set of 3D interaction features and geometry, introducing latent $z$ which, via $p(x \mid c) = \int p_\theta(x \mid c, z)p(z)\,dz$, models the many-to-many relationship between pharmacophores and ligands. **PGMG** (Zhu et al., 2023) represents a pharmacophore as a fully connected graph, encodes it with a GNN, and uses a Transformer decoder to generate SMILES; stereochemistry tokens are omitted since the pharmacophore graph lacks stereo information. **GCPG** (Zou et al., 2025) is a Transformer encoder–decoder whose hidden state is modulated by gating on pharmacophore embeddings and user-set targets to property-controlled sampling.

**Diffusion models** Diffusion models are trained to approximate the reverse process of a predefined forward noising process (Ho et al., 2020). **E(3)DM** (Hoogeboom et al., 2022) is an E(3)-equivariant model that jointly denoise atom coordinates and types. **DiffSBDD** (Schneuing et al., 2024) is an SE(3)-equivariant 3D-conditional model that processes both atomic coordinates and categorical atom features while conditioning on the protein pocket. **TargetDiff** (Guan et al., 2023) conditions the diffusion process on a protein binding site (SE(3)-equivariant), generating ligand coordinates and atom types. Beyond 3D structure, **TGM-DLM** (Gong et al., 2024) denoises token embeddings with non-target specific prompts and post-hoc validity repair. **ProtoBind-Diff** (Mistryukova et al., 2025) is a structure-free diffusion language model, that takes a protein's amino-acid sequence, and generates target-specific ligand candidates.

**Flow matching** Flow matching models learn continuous-time velocity fields transporting a base distribution to the data distribution. **DrugFlow** (Schneuing et al., 2025) is a pocket-conditioned ligand generation model with flow-based sampling.

## 3 BENCHMARK CONSTRUCTION

We evaluate three generator families - *unconditional*, *ligand-based*, and *protein-based* - under a unified, reproducible Five-Stage Filtering Pipeline (Fig. 1). The pipeline thresholds and processing are target-agnostic; for structure-based stages we instantiate experiments on KRAS G12D (switch-II pocket; PDB ID: pdb_00007ew9).

Before any filtering, we standardize molecules with RDKit (Landrum, 2013) and Dimorphite-DL (Ropp et al., 2019). As part of the preparation stage, duplicate molecules were removed within each model's generation set, while duplicates across different models were retained. Validity is checked with RDKit. Then we do the following steps: (i) remove salts and solvents and keep largest organic fragment; (ii) add hydrogens to complete valences; (iii) normalize valence, kekulize, and sanitize; (iv) generate ionization states at $pH\,7.4 \pm 0.0$; (v) preserve declared stereochemistry and, where unspecified, generate up to 8 stereocenters; (vi) generate 3D conformers via distance geometry, minimize them, and retain the lowest-energy conformer for each state.

Specific algorithms and configurations for molecule preprocessing are described in Appendix A.

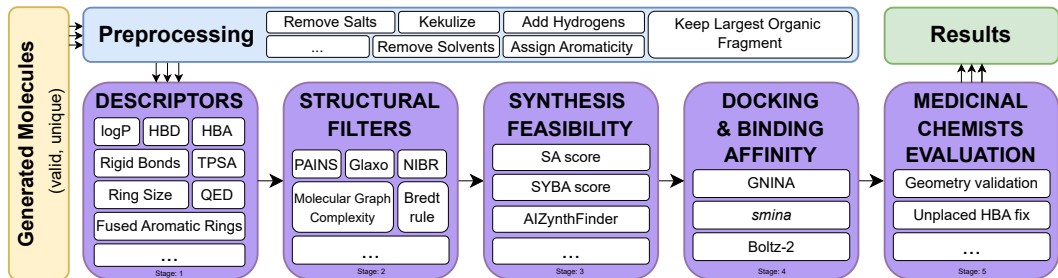

Figure 1: Five-stage filtering pipeline to evaluate generative models. At each stage molecules must satisfy all stage-specific thresholds to proceed. Starting from valid, unique generated molecules, we (i) standardize and generate chemically relevant microstates; (ii) apply physicochemical descriptor filtering; (iii) screen structural and medicinal chemistry alerts; (iv) assess synthesis feasibility and require at least one AiZynthFinder route; (v) evaluate binding compatibility and binding affinity to the KRAS G12D switch-II pocket; (vi) compounds that passed all previous stages receive a blind medicinal chemist review.

## 3.1 STAGE 1: PHYSICOCHEMICAL DESCRIPTORS

We compute 18 two-dimensional physicochemical descriptors (MW, logP, TPSA, HBDs, HBAs, rotatable bonds, number of rings, $f_{sp^3}$, QED, etc.) using RDKit after the preprocessing workflow. Rather than applying any single canonical rule set (e.g., strict Lipinski Rule-of-Five (Lipinski, 2004) or Veber's Rules (Veber et al., 2002)), we combined multiple rule sets and extended thresholds to remove clearly outliers and chemically implausible structures while retaining diversity. This approach reflects the fact that no single rule set covers all descriptors, and our aim of a general, task-agnostic filtration. Appendix B.1 reports exact per-descriptor definitions, bounds, and per-model pass rates.

## 3.2 STAGE 2: STRUCTURAL FILTERS

Molecules that pass the Descriptors stage are further screened with public structural alert sets and graph sanity check to remove reactive, unstable, and toxic molecules: PAINS (Baell & Holloway, 2010), Glaxo (Hann et al., 1999), Inpharmatica (Emmanuel et al., 2025), SureChEMBL (Papadatos et al., 2016); a molecular graph filter (e.g., removal of molecules containing atoms embedded in multiple 3–4-membered rings); a complexity outlier filter (e.g., Bertz (Bertz, 1981), Whitlock (Whitlock, 1998), SMCM (Allu & Oprea, 2005), TWC (Gutman et al., 2001)); the Novartis hit-triage (NIBR) filter (Schuffenhauer et al., 2020); and a Bredt-rule check (Fawcett, 1950). Implementation details, rule lists, and alert sets samples are provided in Appendix B.2.

## 3.3 STAGE 3: SYNTHESIS FEASIBILITY

We assessed synthesizability with three independent predictors: Synthetic Accessibility score (SA score) (Ertl & Schuffenhauer, 2009), Retrosynthetic Accessibility score (RA score) (Thakkar et al., 2021), and Synthetic Bayesian Accessibility (SYBA) score (Voršilák et al., 2020). SA score combines fragment frequencies (from PubChem (Kim et al., 2023)) to yield a synthetic complexity score from 1 (easy) to 10 (hard). RA score is a classifier predicting probability of being a synthetic path for a compound. SYBA is a fragment-based Bernoulli Naive Bayes classifier - trained on ZINC15 "easy" and Nonpher-generated "hard" molecular sets - that classifies structures as easy or hard to synthesize.

We then attempt route finding with AiZynthFinder (Genheden et al., 2020). AiZynthFinder is a machine-learning-guided retrosynthetic workflow. It performs Monte Carlo Tree Search guided by a neural network policy over reaction templates (extracted from USPTO and applied with RDChiral), stopping when all precursors are in stock or the search depth is exceeded. Each compound was processed independently with a maximum reaction depth of 5 steps and tree search budget of 300 s

per molecule. Implementation details - e.g., exact SA score, RA score, SYBA score threshold, and AiZynthFinder configuration - are provided in Appendix B.3.

### 3.4 STAGE 4: DOCKING SCORE AND BINDING AFFINITY ESTIMATION

We dock all molecules that passed previous stages into the KRAS G12D switch-II pocket (PDB ID: pdb_00007ew9). Before docking, the protein structure was prepared by removing water molecules and ligands, and by adding hydrogens and charges using AutoDockTools (Forli et al., 2016). Molecular docking was performed using *smina* (Koes et al., 2013), and GNINA (McNutt et al., 2021). We estimated target binding affinity with deep learning approach Boltz-2 (Passaro et al., 2025), with a $100\mu$M threshold.

A ligand passes Stage 4 if its best docking score is not higher than $\tau_{\text{dock}} = -6.5$ kcal/mol in both engines and binding affinity score is less than $100\mu$M. Docking parameters are detailed in Appendix B.4.

### 3.5 STAGE 5: MEDICINAL CHEMISTS EVALUATION

Molecules that passed all previous stages are scored by a senior medicinal chemist, blinded to model identity. We used PoseBusters (Buttenschoen et al., 2024), RDKit (Landrum, 2013), ProLIF (Bouysset & Fiorucci, 2021). The evaluation follows five criteria designed to capture general medicinal chemistry principles and target-specific knowledge:

- (i) Pose validation by geometry using PoseBusters: molecules exhibiting unnatural torsions, distorted bond angles, or severe intramolecular and intermolecular clashes were excluded.
- (ii) Pose validation by conformational energy using PoseBusters: docking programs frequently place ligands in energetically unfavorable conformations in order to maximize local protein–ligand interactions. If the docked pose was substantially higher in energy than alternative conformers, it was deemed unlikely to represent a realistic binding mode and the molecule was deprioritized.
- (iii) Hydrogen bond donors and acceptors using ProLIF and RDKit: unoccupied hydrogen bond donors (HBDs) and acceptors (HBAs) are penalized, as polar groups are energetically favored to remain solvent-exposed. Their presence in a buried pocket is only justified if supported by strong interactions. Particular attention was given to HBDs, whose number is more stringently limited in drug-like compounds, whereas HBAs can be somewhat more tolerated.
- (iv) Pocket burial using RDKit: to ensure that the ligand fits entirely within the binding pocket rather than protruding into solvent, the maximum distance of any ligand atom to the nearest protein atom was measured. Molecules with atoms extending farther than 5 Å were discarded.
- (v) Target-specific interaction with Asp12 using ProLIF: selectivity for KRAS G12D over wild-type KRAS critically depends on interactions with Asp12. Molecules failing to engage Asp12 were deprioritized, as their likelihood of selective binding was considered low.

## 4 RESULTS

We compare three generator families with six unconditional generators, seven ligand-based generators, and nine protein-based generators. For each model we sample $N_{\text{gen}} = 10,000$ molecules. Validity is checked with RDKit; invalid samples are discarded and resampled; duplicates are removed and resampled within each model's batch. This yields 60,000 molecules for unconditional models, 70,000 molecules for ligand-based models, and 80,000 molecules for protein-based models. Table 2, Table 3, and Table 4 report cumulative pass rates after each step of the pipeline for each model.

We frame a set of architecture-informed hypotheses and then test them under the Pipeline. Equivariant diffusion models explicitly model 3D coordinates and Euclidean symmetries, so they are expected to produce geometrically plausible ligand poses and improved docking performance

(E(3)DM, DiffSBDD, TargetDiff). Graph-based VAEs with scaffold-aware decoders have shown to yield high validity, but may sample synthetically complex chemotypes that are hard to synthesize without additional constraints (JT-VAE, HierGraphVAE, MoLeR). Autoregressive SMILES models are highly sensitive to the learned prior, which substantially alter novelty, similarity to training set, and downstream filtering rates (REINVENT4). Genetic optimizers can rapidly find high scoring and novel molecules without pretraining but may increase structural-alert incidence and reduce synthetic success (MolFinder). Flow matching approaches have reported stable training and efficient sampling that preserves training distribution fidelity (DrugFlow). Pharmacophore-guided methods explicitly bias generation toward interaction motifs and therefore are expected to increase docking enrichment (GCPG, PGMG). Finally, prior work has repeatedly shown that high performance on common generative benchmarks does not guarantee synthesizability in practice, motivating our explicit retrosynthesis and AiZynthFinder gate.

Overall pass rate is low for all families: 364 molecules ($0.607\%$ of 60,000) from unconditional generators, 287 molecules ($0.41\%$ of 70,000) from ligand-based generators, and 318 molecules ($0.398\%$ of 90,000) from protein-based generators. Unconditional models are the most successful, with $0.607\%$ from initial number of molecules passing all stages, showing that such models are able to capture general molecular constraints much better than conditioned models. This may be due to overfitting to features that do not translate into tractable, candidates acceptable for medicinal chemistry.

Figure 2a shows the top three molecules from each model family. Consensus scores were calculated as the arithmetic mean of the inverted and min-max normalized values of *smina* and GNINA docking scores, and the Boltz-2 binding affinity predictions. Figure 2b shows some synthesis path calculated via AiZynthFinder tool for the top molecules. Rest paths are available in the Appendix B.3.

(a) Top generated molecules among three families. Top: unconditional generators (015 - MolGPT, 012 - JT-VAE), middle: protein-based generators (008 - DrugFlow), bottom: ligand-based generators (016 - GCPG).

(b) Synthesis paths for some of the top molecules. Top: MolGPT 015-07075 molecule; middle: DrugFlow 008-03063 molecule; bottom: GCPG 016-07684 molecule.

Figure 2: The top nine generated molecules with their synthesis paths.

## 4.1 Unconditional Molecule Generators

We evaluate E(3)DM, HierGraphVAE, JT-VAE, MoLeR, MolGPT, and TGM-DLM. These models do not condition on target ligands or pocket structure. Results are presented in Table 2. VAE models (especially JT-VAE) retain markedly more candidates through structural filters and synthetic accessibility estimation stages than E(3)DM or MolGPT models, showing that these models are able to sample molecules that are valid and not chemically complex; E(3)DM collapses at synthetic acces-

sibility stage and no candidates remain after this stage; TGM-DLM leaves with the least candidates, mostly due to struggle with validity, filtering out most molecules on descriptors stage.

Table 2: Comparison of unconditional models, each with initial number of molecules $N_{\text{gen}} = 10,000$

| Stage /Model | E(3)DM | HierGraphVAE | JT-VAE | MoLeR | MolGPT | TGM-DLM |
|---|---|---|---|---|---|---|
| Descriptors | 3520 | 3579 | **7586** | 3193 | 3474 | 1216 |
| Structural Filters | 75 | 1176 | **2765** | 718 | 1029 | 100 |
| Synthesis Feasibility | 0 | 975 | **1549** | 557 | 679 | 35 |
| Docking & Binding Aff. | 0 | 477 | **816** | 323 | 340 | 10 |
| Med.Chem. Evaluation | 0 | 53 | **181** | 65 | 64 | 1 |
| **Pass** | 0 | **53** | **181** | **65** | **64** | **1** |

## 4.2 Ligand-based Molecule Generators

For benchmarking, we compare baselines: GCPG, GENTRL, MolFinder, PGMG, and three different setups of REINVENT4: REINVENT4 (V), REINVENT4 (P), REINVENT4 (TL) described below. We examine multiple REINVENT4 setups because sampling behavior depends strongly on the learned prior and fine-tuning strategy; comparing variants isolates how prior choice and transfer learning affect diversity, novelty, synthesizability, and downstream performance.

REINVENT4 (V) (vanilla) uses the out-of-the-box prior released by the authors, and no further modifications applied to the model.

REINVENT4 (P) (prior) is a similarity-based REINVENT4 prior released by the authors that was trained under a medium Tanimoto similarity sampling mode.

REINVENT4 (TL) (transfer learning) is our transfer-learned prior, fine-tuned on known KRAS G12D inhibitors to bias sampling toward the target chemical space. Training and implementation details are provided in Appendix C.

Results are presented in Table 3. GCPG sustains the highest end-to-end retention, resulting in 110 molecules after medicinal chemists evaluation stage. REINVENT4 (V) yields more success molecules (93) than REINVENT4 (P) and REINVENT4 (TL) with 17 and 32 molecules respectively, showing that a broader prior favored downstream filtering pipeline, although sampling 10,000 molecules for REINVENT4 (V) required more attempts. PGMG underperforms early at the descriptors stage and retains the fewest candidates, however, the fraction of molecules that passed docking and binding affinity estimation stage with respect to synthetic feasibility stage is the highest ($19/22 = 0.864$), indicating that pharmacophore-based models tend to generate molecules, that are indeed likely to capture pocket shape geometry and complementarity, but PGMG struggles with overall molecule validity.

Table 3: Comparison of ligand-based models, each with initial number of $N_{\text{gen}} = 10,000$ molecules

| Stage /Model | GCPG | GENTRL | MolFinder | PGMG | REINVENT4 (V) | REINVENT4 (P) | REINVENT4 (TL) |
|---|---|---|---|---|---|---|---|
| Descriptors | **6616** | 5669 | 1592 | 195 | 4089 | 936 | 1204 |
| Structural Filters | **4168** | 1925 | 366 | 37 | 1325 | 593 | 413 |
| Synthesis Feasibility | **1064** | 303 | 265 | 22 | 918 | 222 | 276 |
| Docking & Binding Aff. | **648** | 238 | 200 | 19 | 518 | 72 | 164 |
| Med.Chem. Evaluation | **110** | 24 | 7 | 4 | 93 | 17 | 32 |
| **Pass** | **110** | 24 | **7** | **4** | **93** | **17** | **32** |

## 4.3 Protein-based Molecule Generators

For benchmarking, we compare baselines: DiffSBDD, Dragonfly, Dragonfly biased (b), DrugFlow, Pocket2Mol, ResGen, TargetDiff. We evaluated two different Dragonfly setups to investigate the overall performance of an unmodified model provided by the authors, and a fine-tuned model biased with only one target compound descriptors.

Dragonfly is an out-of-the-box model released by the authors with no modifications applied to the model.

Dragonfly (b) (biased) leverages built-in ability to condition sampling on target compound descriptors. Specifically, bias is applied toward molecular weight, number of rotatable bonds, hydrogen bond donors and acceptors, topological polar surface area, and logP, thereby steering the generation toward molecules with physicochemical properties aligned with the target profile.

Results are presented in Table 4. Although Dragonfly passes the first stage with only 27.79% of initial molecules, the number of molecules that pass medicinal chemists evaluation is the highest among all families, suggesting that Dragonfly is able to sample molecules that are likely to be valid and useful. DiffSBDD, Dragonfly, DrugFlow, Pocket2Mol and TargetDiff strongly dominate other models while passing descriptors stage. However, DiffSBDD, DrugFlow, Pocket2Mol, TargetDiff loses more than a quarter of molecules after structural filters stage, suggesting that those models struggle with synthesis of non toxic and pan-assay-free molecules. TargetDiff does not pass docking and binding affinity estimation stage, and DiffSBDD does not pass medicinal chemistry evaluation stage, while DrugFlow is the second most successful model.

Table 4: Comparison of protein-based models, each with initial number of molecules $N_{\text{gen}} = 10,000$

| Stage /Model | DIFFSBDD | DRAGONFLY | DRAGONFLY (B) | DRUGFLOW | POCKET2MOL | PROTOBIND-DIFF | RESGEN | TARGETDIFF |
|---|---|---|---|---|---|---|---|---|
| Descriptors | 3665 | 2779 | 1022 | 5464 | 2657 | 1466 | 1080 | 3444 |
| Structural Filters | 197 | 1459 | 218 | 1392 | 682 | 195 | 255 | 136 |
| Synthesis Feasibility | 24 | 1207 | 38 | 453 | 137 | 102 | 62 | 4 |
| Docking & Binding Aff. | 13 | 575 | 15 | 344 | 69 | 66 | 37 | 0 |
| Med.Chem. Evaluation | 0 | 227 | 4 | 62 | 12 | 7 | 6 | 0 |
| **Pass** | **0** | **227** | **4** | **62** | **12** | **7** | **6** | **0** |

## 5 DISCUSSION AND CONCLUSION

Applying the same five-stage filtration across unconditional, ligand-based, and protein-based models reveals that across 210,000 generated molecules only 969 (0.461%) of generated molecules pass end-to-end screening. Empirically, unconditional models have the highest overall pass rate of 0.607%, producing molecules that correlate with basic requirements of early drug discovery. Ligand-based models achieved moderate retention with 0.41% pass rate. Protein-based models are left with the smallest fraction of passed molecules (0.398%), with Dragonfly achieving the highest final pass rate of 227 molecules despite low initial retention. Across all families, streepest attrition occurs at synthetic feasibility ($\approx 0.2501$ molecules w.r.t. descriptors filtration) and medicinal chemistry ($\approx 0.1535$ molecules w.r.t. synthesis feasibility evaluation stage) evaluation stages. This confirms prior findings that benchmark metrics, such as validity is weak predictor of downstream utility. The explicit retrosynthesis gate (AiZynthFinder) is therefore critical to separate benchmark overfitting from true drug-likeness.

Our results highlight several architecture-dependent trends across molecule generators. Equivariant diffusion models (E(3)DM, DiffSBDD, TargetDiff) exceed at encoding 3D symmetries and geometric constraints, yet collapse under synthetic accessibility evaluation, suggesting that geometric fidelity alone is insufficient for practical usage. Graph-based VAEs (JT-VAE, HierGraphVAE) balance validity and synthesizability better than other unconditional models, confirming the hypothesis that scaffold-aware decoders reduce chemical complexity. REINVENT4 is highly sensitive to prior choice: broad priors generalize well through the pipeline, while similarity-based or transfer-learned priors reduce downstream retention. Genetic optimizers (MolFinder) find high-scoring candidates without pretraining but enrich high structural alert rates, highlighting the exploration-safety trade-off. Pharmacophore-based models (GCPG, PGMG) confirm the value of explicit interaction motif bias, yielding high enrichment, although PGMG shows low overall pass-rate.

Our findings emphasize that standard generative benchmarks are not good proxies for real-world performance. Optimizing for validity, synthesis, or pocket fidelity independently is insufficient for actionable chemical space that requires alignment across all objectives simultaneously. That is why evaluation should integrate: (i) multistage filtering pipeline (descriptors, structural alerts, synthesis feasibility, docking and binding affinity, and medicinal chemistry stages); (ii) synthesis-aware metrics beyond SA scores; and (iii) stage failures analysis.

While our pipeline integrates synthesis and docking gates, it does not yet capture long-range pharmacokinetics, ADMET liabilities, or clinical viability. Future work should couple generative models with multiscale predictions, and uncertainty-aware evaluation of generated molecules.

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
