# OpenReview forum: "When Validity Isn't Enough: Reliability Gaps in Molecular Generation and KRAS Case Study"
_ICLR.cc/2026/Conference — ICLR 2026 Conference Desk Rejected Submission_

### Official Review · Reviewer_acdA · 2025-10-28

**Soundness:** 2
**Presentation:** 2
**Contribution:** 2
**Rating:** 2
**Confidence:** 4

**Summary:**

This paper addresses a critical disconnect in molecular generation research: modern generative models often achieve high scores on standard metrics (e.g., validity, diversity) but fail to produce molecules viable for real-world drug discovery (e.g., synthesizable, target-binding). To bridge this gap, the authors propose the Five-Stage Filtering Pipeline—a target-agnostic, practice-oriented benchmark—and apply it to 18 molecular generators across three families (unconditional, ligand-based, protein-based), with a focus on KRAS G12D (a mutant lacking approved inhibitors) as a case study. Experiments generated 210,000 molecules and found that less than 1% passed all stages

**Strengths:**

1. Comprehensive model coverage
The study includes 18 generators across architectures (VAEs, autoregressive models, diffusion models, genetic algorithms) and families, enabling fair cross-family comparisons. For example, it tests variants like REINVENT4 (V/P/TL) to isolate prior choice effects, and Dragonfly (biased/unbiased) to assess descriptor conditioning.
2. Physics-informed validation
The KRAS G12D case study is biologically meaningful (no approved inhibitors) and uses a well-characterized pocket (PDB 7EW9). Medicinal chemist review includes target-specific checks (e.g., Asp12 interactions for KRAS selectivity), grounding results in real drug design priorities.

**Weaknesses:**

1.  Scalability and System Size Limitations
The pipeline and experiments focus on small-to-medium molecular systems but ignore large biomolecular systems (e.g., proteins with millions of atoms, multi-component complexes). This limits relevance for drug discovery scenarios like protein–protein interaction inhibitors or bulk material-based therapeutics. Additionally, the paper does not evaluate how the pipeline scales with increasing molecular size.
2. Limited target diversity
The case study focuses exclusively on KRAS G12D. While this is a high-priority target, the pipeline’s "target-agnostic" claim is not fully validated without testing on other targets (e.g., EGFR, CDK2) with different pocket geometries or ligand binding modes. This raises questions about whether the observed pass rate trends (e.g., unconditional model superiority) generalize to other biological targets.
3. Limited insight into failure mechanisms
While the paper identifies high attrition at synthesis feasibility and medicinal review, it does not deeply analyze why models fail at these stages.

**Questions:**

The authors should seriously address the concerns shown in Weakness point by point to improve the quality of the paper.

---

### Official Review · Reviewer_c6eY · 2025-10-31

**Soundness:** 2
**Presentation:** 4
**Contribution:** 1
**Rating:** 2
**Confidence:** 5

**Summary:**

The work introduces the Five-Stage Filtering Pipeline, which streamlines evaluating molecular designs. The proposed setup comprises filters on molecular descriptors, structure, synthetic accessibility, docking score, and medicinal chemist feedback. Unconditional, ligand-based, and structure-based generators with different molecule representations are evaluated within the framework for KRAS binding. The main finding of the study is that unconditional generators can more often pass the quality filters than conditional ones, and ligand-based generators tend to outperform structure-based models.

While the study contains some interesting outcomes, I am recommending rejecting the current version for the following reasons:
1. The proposed evaluation framework is a combination of popularly used medicinal chemistry filters, not offering a new aspect to molecular design evaluation.
2. Only one design task is studied.
3. The reasons behind the findings are unclear.

**Strengths:**

1. A broad set of models is tested, ranging from unconditional to structure-based generators, and from string-based to 3D generators.
2. A multi-faceted evaluation is performed, and a broad range of molecular quality is considered.
3. The paper is well-written and easy to follow.
4. The proposed filters are coming from practical medicinal chemistry considerations, with a strong know-how.

**Weaknesses:**

1. The proposed framework unifies known medicinal chemistry filters in a standard way. Even the popularly adopted library REINVENT implements and adopts some of these filters and contains scoring function implementations. While making these filters a core part of a benchmark is an interesting idea, the work studies only one design task and falls short of being comprehensive as a benchmark. Moreover, the transferability of the model comparisons to other targets remains only speculative.
2. The work misses the opportunity to provide actionable insights and guidelines for future molecular generation model development. The reasons behind the model performances are not investigated. Otherwise, the work remains purely critical.
3. The impact of the (pre)training data is never considered during evaluation, which can have a substantial impact on outcomes for unconditional models. It is difficult to understand *why* some models outperform others without considering the training data. For instance, do unconditional string-based models outperform 3D structure-based models because of the model architecture/representation, or because of the larger training data they adopt? I recommend training (some) models of each category with the same training data and repeating the analysis.
4. The benchmark misses the diversity aspect of evaluation, which is key to molecule design. This should be included in future versions of the benchmark.
5. While medicinal chemist feedback is important, it is inaccessible to future researchers who want to benchmark their models. Can this part be replaced with a model, e.g., a classifier trained on medicinal chemist feedback data (e.g., www.nature.com/articles/s41467-023-42242-1)? Also, having multiple medicinal chemists in this study could increase the reliability of the current assessments, since chemists can also disagree (www.nature.com/articles/s41467-023-42242-1).

A friendly suggestion (not a weakness): Presenting the tool as a web server/API where people can upload their generated molecules and download the scores would significantly increase the usability and impact of the work. Setting up all the tools used here can be cumbersome for many users.

**Questions:**

1. What is the added value of the work, beyond unifying known medicinal chemistry filters into a single evaluation template?
2. How generalizable are the findings to other protein targets?
3. How do authors explain unconditional models - which have no access to binding data - outperforming conditional models, in a complex task such as bioactive molecule design?

---

### Official Review · Reviewer_zYP8 · 2025-11-01

**Soundness:** 2
**Presentation:** 2
**Contribution:** 2
**Rating:** 2
**Confidence:** 4

**Summary:**

This paper presents a 5 stage pipeline for evaluating molecular generators.

**Strengths:**

The paper is relatively clear and easy to understand.

**Weaknesses:**

The main weakness of this paper is that the contribution and innovation seem weak. The checks that are performed are not novel in themselves, and not surprising. It’s not clear that the ordering is important. If the point is that few molecules can go on to be drugs, I think that is already accepted in the community.

**Questions:**

One of the steps is to remove salts and solvents. Why are generators generating those, especially the conditional ones? Is it that they are generating lots of junk?

Why is “keeping the largest fragment” a step? Are these generators generating multiple fragments in a single generation step?

What exactly were the unconditional generators doing? By unconditional here, I assume that they were not aware of the target? If so, wouldn’t their molecules basically be random? If so, was the binding affinity just due to randomness?

What about off-target binding?

---

### Official Review · Reviewer_oUBS · 2025-11-01

**Soundness:** 3
**Presentation:** 3
**Contribution:** 2
**Rating:** 2
**Confidence:** 4

**Summary:**

The paper presents an intriguing study on evaluating generative models for molecular design. The proposed benchmark includes five stages: descriptors-based filtering, structural filters, synthesizability, molecular docking, and medicinal chemist evaluation. Each stage incorporates multiple filters and criteria to ensure only the most promising compounds advance to the next stage. Descriptor-based filters are based on various rule sets, such as the Lipinski Rule of Five. Structural filters rely on sets of structural alerts. Synthesizability is evaluated through multiple independent methods, including the prediction of synthesis routes. Docking is conducted using AutoDock Vina. The final stage involves manual inspection of binding poses by experienced senior medicinal chemists. The benchmark is prepopulated with different classes of generative models. The results lead to interesting conclusions, notably the superior performance of unconditional generators in passing all filters.

**Strengths:**

- The proposed benchmark provides a comprehensive set of evaluation criteria, offering a multi-perspective view of the generated compounds.
- Different classes of generative models are benchmarked, and the differences between their performance are discussed.
- This paper offers valuable insights for practitioners who want to apply generative models in their drug discovery campaigns.
- All generated molecules are preprocessed in the same way to produce comparable results.
- The paper is written in a very clear manner, with all filters and property ranges provided in the appendix.

**Weaknesses:**

- In the first paragraph of the introduction, early drug discovery might be somewhat misrepresented. Specifically, the binding pocket is sometimes unknown and not always necessary for discovering a new active molecule. In fact, sometimes the binding pocket is discovered after identifying a binding compound, as a result of the co-crystallization of the ligand with the protein. Furthermore, crystallography and pocket prediction are not the only methods used to identify a pocket; it can also be, for example, a Cryo-EM structure or techniques like HDX-MS.
- The text says that thresholds for rule sets were extended “to remove clearly outliers.” What criteria were used to determine how much these thresholds should be extended? Was it based on some outlier detection method?
- The benchmark is limited to only one biological target, so it is unclear if the conclusions would be the same for other targets.
- The value of this benchmark for the machine learning community is unclear. Undoubtedly, this strict evaluation protocol can have a significant impact on drug discovery projects; however, its applicability to evaluating new generative methods is limited due to the manual nature of the final evaluation stage. Perhaps this paper would be a valuable review of the current state of the field of generative molecular design as a journal paper if it included more models. As a benchmark paper, the last manual stage renders this benchmark practically unusable. Moreover, some criteria may be too stringent as current approved drugs are more and more often beyond the Rule of Five, and synthetic routes predicted by a model can be incorrect or missing available starting materials.
- Adding more details on the expert evaluation would make this pipeline more reproducible, especially since the paper is still one page below the page limit. The criteria in Section 3.5 seem to be fairly easy to automate. How are these criteria combined with the chemists’ scores? Are they just a summary of what chemists assessed, or are they computed and given to the chemists for evaluation?
- The code repository linked in the appendix does not work (the content cannot be found). For this benchmark's success, it will be crucial to provide a high-quality code repository that automates the evaluation process. Providing an easy-to-use evaluation code and all necessary data would convince me to increase my score.
- The emphasis on this benchmark being the first of its kind seems somewhat misleading. These filters, used in various combinations, have also been employed in other benchmarks and practical screening pipelines. Even if this particular combination and exhaustiveness are unique, each step is a well-known and commonly used procedure.

**Questions:**

1. Did you test the variability of chemists’ scores? Do you think this manual evaluation can be reproduced by another chemist? What is the minimum level of expertise required to accurately score the generated compounds?
2. Do you think that the lack of any approved inhibitors for this target impacts the quality of the models, especially Boltz-2? Do you think that the model may be biased towards other mutants for which approved drugs exist?

---

### Note · Program_Chairs · 2026-01-17
**Submission Desk Rejected by Program Chairs**

The following references in this submission do not refer to real documents and/or have major errors in bibliographic information:

 Olivier Bodenreider et al. Artificial benchmarks for molecular generation do not correlate with practical drug discovery performance. Drug Discovery Today, 26(8):1863–1870, 2021.

Gisbert Schneider and Uli Fechner. Lead- and drug-like compounds: the role of structural complex-ity and synthetic accessibility in drug discovery. Nature Reviews Drug Discovery, 9(12):949–962, 2010.